# Molecular Manipulation of the miR160/*AUXIN RESPONSE FACTOR* Expression Module Impacts Root Development in *Arabidopsis thaliana*

**DOI:** 10.3390/genes15081042

**Published:** 2024-08-07

**Authors:** Kim Zimmerman, Joseph L. Pegler, Jackson M. J. Oultram, David A. Collings, Ming-Bo Wang, Christopher P. L. Grof, Andrew L. Eamens

**Affiliations:** 1Centre for Plant Science, School of Environmental and Life Sciences, College of Engineering, Science and Environment University of Newcastle, Callaghan, NSW 2308, Australia; kim.zimmerman@uon.edu.au (K.Z.); joseph.pegler@newcastle.edu.au (J.L.P.); jackson.oultram@uon.edu.au (J.M.J.O.); david.collings@anu.edu.au (D.A.C.); chris.grof@newcastle.edu.au (C.P.L.G.); 2Research School of Biology, Australian National University, Canberra, ACT 2601, Australia; 3CSIRO Agriculture and Food, Canberra, ACT 2601, Australia; ming-bo.wang@csiro.au; 4School of Agriculture and Food Sustainability, The University of Queensland, St Lucia, QLD 4072, Australia; 5Seaweed Research Group, School of Health, University of the Sunshine Coast, Maroochydore, QLD 4558, Australia

**Keywords:** *Arabidopsis thaliana* (*Arabidopsis*), root development, microRNA (miRNA), miR160, miR160 expression module, miR160-directed gene expression regulation, *AUXIN RESPONSE FACTOR* (*ARF*), *ARF10*, *ARF16*, *ARF17*, molecular manipulation

## Abstract

In *Arabidopsis thaliana* (*Arabidopsis*), microRNA160 (miR160) regulates the expression of *AUXIN RESPONSE FACTOR10* (*ARF10*), *ARF16* and *ARF17* throughout development, including the development of the root system. We have previously shown that in addition to DOUBLE-STRANDED RNA BINDING1 (DRB1), DRB2 is also involved in controlling the rate of production of specific miRNA cohorts in the tissues where *DRB2* is expressed in wild-type *Arabidopsis* plants. In this study, a miR160 overexpression transgene (*MIR160B*) and miR160-resistant transgene versions of *ARF10* and *ARF16* (*mARF10* and *mARF16*) were introduced into wild-type *Arabidopsis* plants and the *drb1* and *drb2* single mutants to determine the degree of requirement of DRB2 to regulate the miR160 expression module as part of root development. Via this molecular modification approach, we show that in addition to DRB1, DRB2 is required to regulate the level of miR160 production from its precursor transcripts in *Arabidopsis* roots. Furthermore, we go on to correlate the altered abundance of miR160 or its *ARF10*, *ARF16* and *ARF17* target genes in the generated series of transformant lines with the enhanced development of the root system displayed by these plant lines. More specifically, promotion of primary root elongation likely stemmed from enhancement of miR160-directed *ARF17* expression repression, while the promotion of lateral and adventitious root formation was the result of an elevated degree of miR160-directed regulation of *ARF17* expression, and to a lesser degree, *ARF10* and *ARF16* expression. Taken together, the results presented in this study identify the requirement of the functional interplay between DRB1 and DRB2 to tightly control the rate of miR160 production, to in turn ensure the appropriate degree of miR160-directed *ARF10*, *ARF16* and *ARF17* gene expression regulation as part of normal root system development in *Arabidopsis*.

## 1. Introduction

In plants, small regulatory RNAs (sRNAs) of primarily 21 to 24 nucleotides (nt) in length play a key role in controlling gene expression [1,2,3,4,5]. In the genetic model species *Arabidopsis thaliana* (*Arabidopsis*), multiple classes of sRNA are produced with each class generated from a structurally distinct double-stranded RNA (dsRNA) precursor transcript [1,2,3,4,5]. Members of two protein families, including the DICER-LIKE (DCL) and dsRNA BINDING (DRB) protein families, form core functional partnerships to recognize, bind and process each structurally distinct dsRNA precursor transcript to produce each class of sRNA [6,7,8,9]. Upon functional maturity, the sRNA is loaded by a specific protein effector complex, termed the RNA-induced silencing complex (RISC), which harbors a member of the ARGONAUTE (AGO) protein family at its functional core [10,11]. RISC uses the loaded sRNA as a sequence specificity guide to direct gene expression regulation of cognate target genes at either the transcriptional or posttranscriptional level [12,13].

The microRNA (miRNA) class of sRNA is the most thoroughly characterized class of sRNA in *Arabidopsis,* with many of the miRNAs, which accumulate in *Arabidopsis* cells, regulating the expression of developmentally important genes [6,8,12]. Each *Arabidopsis* miRNA originates from an individual genomic locus, termed a *MICRORNA* (*MIR*) gene [14,15,16], which, like protein-coding genes, are transcribed by RNA polymerase II (Pol II) [17,18]. The Pol II transcription product from a *MIR* gene, the primary miRNA (pri-miRNA), contains a region of partial self-complementarity which forms a stem-loop structure of imperfectly paired dsRNA. In the *Arabidopsis* cell nucleus, pri-miRNAs are recognized and bound by SERRATE1 (SE1), a zinc finger protein with the capacity to bind stem-loop-structured RNAs [19]. SE1 transports the bound pri-miRNA to specialized nuclear bodies, called nuclear Dicing bodies (D-bodies), where the pri-miRNA is further bound by DRB1, the functional partner of DCL1 [20,21,22]. DRB1 correctly positions DCL1 on the dsRNA stem-loop region of the pri-miRNA to direct accurate and efficient DCL1-catalyzed pri-miRNA processing to produce the processing intermediate, the precursor-miRNA (pre-miRNA) [6,8,23]. Together with DRB1, DCL1 further processes the pre-miRNA to liberate the miRNA/miRNA* duplex [6,8,23]. The 2′ hydroxyl group at the 3′ terminal nt of each duplex strand is then methylated by the sRNA-specific methyltransferase, HUA ENHANCER1 (HEN1), to stabilize the miRNA guide strand post its separation from the miRNA* passenger strand [24,25]. The miRNA guide strand is preferentially selected over the miRNA* passenger strand by DRB1 for loading into RISC to form the miRISC (miRNA-induced silencing complex) [26]. AGO1 forms the catalytic core of miRISC in *Arabidopsis* and uses the loaded miRNA as a sequence specificity determinant to direct either mRNA cleavage or translational repression to control target-gene expression [12,13,27].

In addition to DRB1, two other DRB protein family members, DRB2 and DRB4, are localized to the nucleus of the *Arabidopsis* cell, a cellular localization which indicates potential roles for DRB2 and DRB4 in miRNA biogenesis which occurs exclusively in the nucleus [9,27,28,29,30,31]. Accordingly, DRB4, together with DCL4, are required to produce the newly evolved subclass of miRNAs from highly structured stem-loop precursors, which form due to the extensive base pairing between the two stem arms, which are separated by a minimal loop region [26,29,32]. Unlike DRB1 and DRB4, which exclusively interact with DCL1 and DCL4, respectively [9,23,29,32], DRB2 is likely able to present miRNA precursor transcripts to both DCLs, forming a functional interaction with DCL1 to produce a specific cohort of conserved miRNAs, or with DCL4 to produce newly evolved miRNAs [29,30,31,32]. These DRB2/DCL interactions appear to interrupt the functionality of the DRB1/DCL1 and DRB4/DCL4 partnerships in the developmentally important tissues of wild-type *Arabidopsis* plants where *DRB2* is expressed [28,29,30,33,34].

*Arabidopsis*, like all other plant species, synthesizes auxin (IAA; indole-3-acetic acid), primarily in the region which surrounds the shoot apical meristem (SAM) [35,36,37]. Auxin is a crucial phytohormone for plant development [37], playing a role in regulating gravitropism [38], phototropism [39], and organ patterning [40], as well as acting as a key regulator of floral development [41] and seed dormancy and/or germination [42,43]. Auxin also forms a key signaling molecule to play a pivotal role in root structure formation, directing cell identity in the root apical meristem (RAM) for the subsequent promotion of primary root elongation and lateral root initiation [44,45]. Auxin continues to accumulate at the RAM of the primary root throughout *Arabidopsis* development to maintain stem cell proliferation in the RAM to ensure the continued elongation of the primary root [45]. In addition, the continued accumulation of auxin throughout root system development directs the formation of ‘*local auxin maxima*’, which form the sites for lateral root initiation via the formation of localized populations of undifferentiated cells [44,46]. After lateral root initiation, the level of auxin remains constant at these sites to ensure continued growth and elongation of lateral roots [44,46,47].

At the molecular level, when auxin is perceived in the nucleus, members of the AUXIN RESPONSE FACTOR (ARF) transcription factor family are activated, and once activated, the ARF regulates the expression of sets of *AUXIN RESPONSE GENES* (*ARG*s) via binding the specific motifs in *ARG* promoter regions, termed *AUXIN RESPONSE ELEMENTS* (*ARE*s), to elicit an appropriate response to auxin [48]. In *Arabidopsis*, ARF10, ARF16 and ARF17 form a subclade within the larger multimember *ARF* gene family, with the expression of these three *ARF*s regulated at the posttranscriptional level by miR160 [49,50]. Our previous research [26] on the *drb2345* quadruple mutant showed that miR160 levels were increased considerably in this mutant background where of the five members of the *Arabidopsis* DRB protein family, only DRB1 is functional. This finding strongly suggested that in addition to DRB1 functioning with DCL1, another DRB family member is involved in miR160 production in *Arabidopsis*. In a subsequent study characterizing the *drb235* triple mutant, we identified that DRB2 most likely occupies this additional role in controlling the level of the miR160 sRNA in *Arabidopsis* vegetative tissues [30].

Here, we used a molecular modification approach to further explore the degree to which the interplay between DRB1 and DRB2 is required for miR160 production and miR160-directed target gene expression regulation in the *Arabidopsis* root system. Towards this goal, wild-type *Arabidopsis* plants (ecotype Columbia-0; Col-0) and the *drb1* and *drb2* single mutants were transformed with the miR160 overexpression transgene, *MIR160B*, and miR160-resistant transgene versions of *ARF10* and *ARF16*, *mARF10* and *mARF16*. Molecular profiling of the roots of the *Arabidopsis* lines Col-0, Col-0/*MIR160B*, Col-0/*mARF10*, Col-0/*mARF16*, *drb1*, *drb1*/*MIR160B*, *drb1*/*mARF10*, *drb1*/*mARF16*, *drb2*, *drb2*/*MIR160B*, *drb2*/*mARF10* and *drb2*/*mARF16*, revealed that DRB2 acts both synergistically and antagonistically to DRB1 function to tightly control the rate of miR160 production from its three precursor transcripts, *PRE-MIR160A*, *PRE-MIR160B* and *PRE-MIR160C*. Comparison of the molecular profiles obtained for the transformant series to those constructed for unmodified Col-0, *drb1* and *drb2* plants, further revealed that promotion of primary root elongation was the result of enhanced miR160-directed repression of *ARF17* expression. This profiling exercise also showed that the promotion of lateral root development in individual plant lines of the generated transformant series was the result of altered miR160-directed expression regulation of *ARF17*, and to a lesser degree, *ARF10* and *ARF16* expression regulation. Similarly, altered expression regulation of *ARF10*, *ARF16* and *ARF17* by miR160 was likely responsible for the promotion of adventitious root development observed in individual transformant lines expressing either the *MIR160B*, *mARF10* or *mARF16* transgene. Taken together, we demonstrate the requirement of the interplay between DRB1 and DRB2 to tightly control the rate of miR160 production, to in turn ensure the correct degree of miR160-directed *ARF10*, *ARF16* and *ARF17* gene expression regulation as part of normal root system development in *Arabidopsis*.

## 2. Materials and Methods

### 2.1. Plant Lines, Growth Conditions, and Phenotypic Assessments

For all experimental analyses performed in this study, the *Arabidopsis* Col-0 ecotype was used as the wild-type control. The single mutants *drb1-1* (SALK_064863) and *drb2-1* (GABI_348A09), which harbor T-DNA insertion mutations in the *DRB1* (*AT1G09700*) and *DRB2* (*AT2G28380*) locus, respectively, are also in the Col-0 background and have been described in detail previously [28,29,30]. Col-0, *drb1* and *drb2* plants were also used to introduce the three plant expression vectors generated for use in this study, including the *MIR160B*, *mARF10* and *mARF16* transgenes. A standard *Agrobacterium*-mediated transformation approach was used to introduce the three transgenes into the Col-0, *drb1* and *drb2* backgrounds to produce nine additional *Arabidopsis* plant lines, including the Col-0/*MIR160B*, Col-0/*mARF10*, Col-0/*mARF16*, *drb1*/*MIR160B*, *drb1*/*mARF10*, *drb1*/*mARF16*, *drb2*/*MIR160B*, *drb2*/*mARF10* and *drb2*/*mARF16* transformant lines.

The seeds sourced from Col-0, *drb1* and drb2 plant lines and the plant lines selected to represent the Col-0/*MIR160B*, Col-0/*mARF10*, Col-0/*mARF16*, *drb1*/*MIR160B*, *drb1*/*mARF10*, *drb1*/*mARF16*, *drb2*/*MIR160B*, *drb2*/*mARF10* and *drb2*/*mARF16* transformant populations were surface sterilized via incubation at room temperature for 90 min (min) in a sealed chamber containing chlorine gas. The sterilized seeds were transferred to petri dishes that contained full strength Murashige and Skoog (MS) plant growth medium supplemented with 1.0% (*w*/*v*) sucrose and 0.8% (*w*/*v*) agar. The dishes were sealed with permeable tape and the seeds stratified via incubation for 48 h (h), in the dark, at 4 °C. Following stratification, dishes were transferred to a temperature-controlled growth cabinet (A1000 Growth Chamber, Conviron^®^, Melbourne, Australia) and cultivated for 10 days under standard *Arabidopsis* growth conditions of 16 h of light (100–150 µmol m^−2^ s^−1^)/8 h of dark and a day/night temperature of 22/18 °C. At day 10, six healthy seedlings per *Arabidopsis* line were transferred to a large petri dish that contained fresh plant growth medium, and post sealing with permeable tape, the plates were orientated vertically and cultivated for a further 11-day period in the temperature-controlled growth cabinet under the growth conditions stated above. For each of the twelve *Arabidopsis* lines analyzed in this study, three plates were prepared via this method, with each plate forming a biological replicate for use in subsequent phenotypic and molecular analyses.

At day 21, images of each of the 12 *Arabidopsis* lines under assessment were captured using a Canon IXUS 801S digital camera with a 4.0 cm^2^ red calibration square included in each captured image. Images with the calibration square were loaded into the software, Easy Leaf Area (https://www.quantitative-plant.org/software/easy-leaf-area, accessed on 1 July 2024). The number of green pixels in each image was determined and via the use of the calibration square with the rosette leaf surface area for each plant line calculated in cm^2^. The same set of generated images was subsequently loaded into ImageJ (https://imagej.net/ij/, accessed on 1 July 2024) with length calibration included. In ImageJ, the number of pixels corresponding to 1.0 cm in length was determined and the ‘*free hand*’ tool used to trace the primary root of each 3-week-old vertically grown seedling. The ImageJ software calculated a final measurement in millimeters (mm) for the length of the primary root. The number of lateral roots and adventitious roots was counted manually using the same image set in ImageJ. As stated above, three biological replicates with six seedlings per replicate were used for this analysis.

### 2.2. Construction of the Transgene Vectors MIR160B, mARF10 and mARF16

The miR160-resistant versions of the *ARF10* and *ARF16* transcripts (termed *mARF10* and *mARF16*, respectively) were commercially synthesized by GenScript (www.genscript.com/, accessed on 1 July 2024). In addition to the full-length coding sequence of each *ARF* gene transcript being synthesized, the *mARF10* sequence was designed to harbor a *Sal*I and *Kpn*I restriction site at its 5′ and 3′ terminus, respectively. Similarly, the *mARF16* sequence was designed to be flanked by a 5′ *Eco*RI and 3′ *Sal*I restriction site as part of the synthesized sequence. Following restriction endonuclease cleavage with the appropriate enzyme combinations, the digested *mARF10* and *mARF16* restriction fragments were directionally cloned into the pGEM^®^-T Easy-based vectors (Promega, Sydney, Australia), pGEM-T::*ARF10pro* and pGEM-T::*ARF16pro,* which had been digested with the same restriction enzyme combinations. The shuttle vectors, pGEM-T::*ARF10pro* and pGEM-T::*ARF16pro,* had been previously generated via insertion of PCR amplicons that represented the native promoter regions of the *Arabidopsis ARF10* (*AT2G28350*) and *ARF16* (*AT4G30080*) genes, an approach that enabled the subsequent placement of the *mARF10* and *mARF16* transgene fragment immediately downstream of the native promoter region sequence of each modified *ARF* gene. The pGEM-T::*ARF10pro-mARF10* vector was then digested with restriction endonucleases *Xho*I and *Kpn*I to release the *ARF10pro-mARF10* transgene fragment which was subsequently cloned into the similarly digested pORE1 vector [51] to produce the plant expression vector, *mARF10*. The pGEM-T::*ARF16pro-mARF16* vector was digested with *Xba*I and *Sal*I to liberate the *ARF16pro-mARF16* restriction fragment which was cloned into the similarly digested pORE1 [51] to produce the *mARF16* plant expression vector. The *Arabidopsis* genomic DNA sequence from which *PRE-MIR160B* is transcribed was amplified by a standard PCR approach using a 5′ primer that contained a *Xho*I restriction site and a 3′ primer that included a *Bam*HI restriction site at its 3′ terminus, with the resulting amplicon cloned into the pGEM^®^-T Easy Cloning vector. The resulting vector, pGEM-T::*MIR160B,* was digested with restriction endonucleases *Xho*I and *Bam*HI to liberate the *PRE-MIR160B* sequence which was subsequently cloned into the similarly digested shuttle vector pART7 [52]. The produced vector pART7::*MIR160B* was digested with restriction endonuclease *Not*I and the liberated *35Spro-MIR160B-OcsT* restriction fragment was cloned into the pBART plant expression vector [52] to produce *MIR160B*. All DNA oligonucleotides used to construct the *MIR160B*, *mARF10* and *mARF16* transgenes are provide in Appendix A.

### 2.3. Introduction of the MIR160B, mARF10 and mARF16 Transgenes into Col-0, drb1 and drb2 Plant Lines via Agrobacterium-Mediated Transformation

Post their introduction and sequence confirmation in *Escherichia coli* (*E. coli*; DH5α strain), the plasmids which harbored the *MIR160B*, *mARF10*, and *mARF16* transgenes were extracted from *E. coli* and introduced into *Agrobacterium tumefaciens* (*Agrobacterium*; GV3101 strain) and the resulting *Agrobacterium* cultures were used to transform Col-0, *drb1* and *drb2* plants according to the protocol of [53]. The seeds were harvested from ‘*floral dipped*’ plants and in the T_1_ generation putative transformants were selected on plant growth medium supplemented with the *in planta* selective agent glufosinate ammonium (Sigma Aldrich, Sydney, Australia) at a concentration of 5.0 μg/mL, and 150 μg/mL of Timentin (PhytoTech Labs, Lenexa, KS, USA), which was also included in the growth medium to remove any unwanted residual *Agrobacterium* growth. The transgene copy number and the zygosity of each putative transformant line was determined via a standard PCR-based genotyping approach of resistant seedlings that were germinated and cultivated on *Arabidopsis* growth medium supplemented with 5.0 μg/mL glufosinate ammonium. Also in the T_2_ generation, the ‘*best performing*’ transformant line determined to be homozygous for a single chromosome insertion event was selected to represent the Col-0/*MIR160B*, Col-0/*mARF10*, Col-0/*mARF16*, *drb1*/*MIR160B*, *drb1*/*mARF10*, *drb1*/*mARF16*, *drb2*/*MIR160B*, *drb2*/*mARF10* and *drb2*/*mARF16* transformant populations for subsequent phenotypic and molecular analyses, with all experimentation reported here conducted on the T_3_ generation of plants. All DNA oligonucleotides used for PCR-based genotyping are listed in Appendix A.

### 2.4. RNA Extraction, Complementary DNA Synthesis, and Gene Expression Analysis

For the reported molecular assessments, total RNA was extracted from three biological replicates of pooled root material sampled from six 3-week-old vertically grown seedlings of plant lines Col-0, Col-0/*MIR160B*, Col-0/*mARF10*, Col-0/*mARF16*, *drb1*, *drb1*/*MIR160B*, *drb1*/*mARF10*, *drb1*/*mARF16*, *drb2*, *drb2*/*MIR160B*, *drb2*/*mARF10* and *drb2*/*mARF16,* using TRIzol^TM^ Reagent according to the manufacturer’s protocol (ThermoFisher Scientific, Sydney, Australia). The quality of the extracted total RNA was assessed via standard electrophoretic separation of the nucleic acid on an ethidium bromide-stained 1.2% (*w*/*v*) agarose gel. For each high-quality total RNA preparation, a NanoDrop spectrophotometer (NanoDrop^®^ ND-1000, Thermo Scientific, Sydney, Australia) was subsequently employed to determine total RNA concentration in micrograms per microliter (µg/µL).

For the generation of a global high molecular weight complementary DNA (cDNA) library for gene expression quantification, 5.0 µg of total RNA was digested with 5.0 units (U) of DNase I according to the manufacturer’s instructions (New England Biolabs, Melbourne, Australia). The DNase I-treated total RNA was next purified using a RNeasy Mini Kit (Qiagen, Melbourne, Australia) according to the manufacturer’s protocol, and 1.0 µg of this purified preparation was then used as the template to synthesize cDNA via the use of 1.0 U of the ProtoScript^®^ II Reverse Transcriptase and 2.5 mM of oligo dT_(18)_ according to the manufacturer’s instructions (New England Biolabs, Melbourne, Australia).

A miR160-specific cDNA was synthesized via the treatment of 500 nanograms (ng) of total RNA with 0.2 U of DNase I (New England Biolabs, Melbourne, Australia). Each DNase I-treated total RNA sample was directly used as a template for miR160-specific cDNA synthesis using miRNA-specific stem-loop DNA oligonucleotides (Appendix A) and 1.0 U of ProtoScript^®^ II Reverse Transcriptase (New England Biolabs, Melbourne, Australia). The cycling conditions of (1) 1 cycle of 16 °C for 30 min, (2) 60 cycles of 30 °C for 30 s, 42 °C for 30 s and 50 °C for 2 s, and (3) 1 cycle of 85 °C for 5 min were used for miR160-specific cDNA synthesis. All generated single-stranded cDNAs were subsequently diluted to a working concentration of 50 ng/µL in RNase-free water prior to their use as a template for the quantification of the abundance of either a gene transcript or miR160.

Quantitative reverse transcriptase polymerase chain reaction (RT-qPCR) assessment of gene expression or miR160 abundance used the same cycling conditions of (1) 1 cycle of 95 °C for 10 min, and (2) 45 cycles of 95 °C for 10 s and 60 °C for 15 s. The GoTaq^®^ qPCR Master Mix (Promega, Sydney, Australia) was used as the fluorescent reagent for all performed RT-qPCR experiments. miR160 abundance and gene transcript expression were quantified using the 2^−∆∆CT^ method with the small nucleolar RNA, snoR101, and *ELONGATION FACTOR-1α* (*EF-1α*; *AT5G60390*) used as the respective internal controls to normalize the relative abundance of miR160 or of each assessed transcript. For all RT-qPCR experiments reported here, three biological replicates were used per sample, and three technical replicates were performed per biological replicate. The sequence of each DNA oligonucleotide used in this study either for the synthesis of a miR160-specific cDNA, or to quantify gene transcript abundance via RT-qPCR is provided in Appendix A.

### 2.5. Statistical Analysis

Analytical data from this study were obtained from three biological replicates with each biological replicate consisting of a pooled sample of six individual plants. Statistical analysis was performed using a standard two-tailed *t*-test. Different degrees of statistical significance are represented in Figures 1–4 via the use of asterisks, where * *p* ≤ 0.05, ** *p* ≤ 0.01, *** *p* ≤ 0.001.

## 3. Results

### 3.1. Molecular Profiling of the miR160 Expression Module in Wild-Type Arabidopsis Plants and the drb1 and drb2 Mutants

Phenotypic assessment of the root architecture of 3-week-old wild-type *Arabidopsis* plants revealed Col-0 plants (Figure 1A) to have an average primary root length of 38 mm (mm) (Figure 1B) and to develop an average of 12 lateral roots (Figure 1C) and 1.5 adventitious roots (Figure 1D). The severe phenotypic alteration of the *drb1* mutant throughout all aspects of its vegetative and reproductive development has been reported previously [6]. Consistently, global alteration of root system architecture was observed in 3-week-old *drb1* mutants (Figure 1A), characterized by a 34% reduction to primary root length (25 mm) (Figure 1B), a 66% reduction to lateral root number (Figure 1C), and a 167% increase in adventitious root number (Figure 1D). While mild in comparison to *drb1*, *drb2* plants also displayed a pattern of vegetative and reproductive development distinct to that of wild-type *Arabidopsis* [28,29,30]. Figure 1A clearly shows the enhanced development of the root system of the *drb2* mutant compared to the root system of Col-0 plants. More specifically, 3-week-old *drb2* mutants showed an average primary root length of 81 mm, and developed an average of 21 lateral roots and 2.3 adventitious roots (Figure 1B–D), representing increases of 113%, 75% and 53%, respectively over Col-0. Impaired vegetative development was also observed in the aerial tissues of *drb1* plants (Appendix A), with rosette leaf area reduced by 41% to 1.6 cm^2^ from 2.7 cm^2^ of 3-week-old Col-0 plants (Appendix A). Similar to the root tissue, an enhanced growth status was also observed for the aerial tissues of 3-week-old *drb2* plants, when compared to Col-0 plants, with rosette leaf surface area increased by 52% to 4.1 cm^2^ (Appendix A).

Our previous analyses of 15-day-old Col-0, *drb1* and *drb2* seedlings following a 7-day cadmium stress period [54], strongly suggested that in addition to DRB1, DRB2 also plays a role in the multitiered regulation of the miR160/*ARF10*/*ARF16*/*ARF17* expression module, most likely via its antagonism of the DRB1/DCL1 functional partnership [30,31]. Therefore, we next assessed the expression of the three miR160 target genes in the inflorescence tissues of Col-0, *drb1* and *drb2* plants grown under standard conditions using semi-quantitative RT-PCR. As shown in Appendix A, all three assessed miR160 target genes were upregulated in the inflorescence tissues of the *drb1* mutant (Appendix A). Importantly, the abundance of the *ARF10* and *ARF16* transcripts was elevated to a similar degree in *drb2* inflorescences as in *drb1* inflorescences, whereas the level of the *ARF17* transcript remained unchanged in the *drb2* mutant compared to Col-0 inflorescences (Appendix A). To investigate the antagonistic effect of DRB2 on the DRB1/DCL1 and DRB4/DCL4 functional partnerships [29,30,31], we next quantified the expression level of *DRB1*, *DRB2*, and *DRB4* in the inflorescence tissues of Col-0, *drb1* and *drb2* plants. As shown in Appendix A, the expression of *DRB1* and *DRB4* was elevated by 1.7- and 1.8-fold in the *drb2* mutant background, supporting the potential antagonistic effect of DRB2 function in the correct regulation of all *Arabidopsis* miRNA expression modules.

The miR160/*ARF10*/*ARF16*/*ARF17* expression module was next profiled by RT-qPCR in the roots of 3-week-old Col-0, *drb1* and *drb2* plants. In *Arabidopsis*, the miR160 sRNA is processed from three structurally distinct precursor transcripts, including *PRE-MIR160A*, *PRE-MIR160B* and *PRE-MIR160C* [55]. As shown in Figure 1E–G, the abundance of the three miR160 precursors was elevated by 30.0-, 2.9- and 14.8-fold, respectively, in *drb1* roots. In *drb2* roots, a similar trend of increased transcript abundance was observed for *PRE-MIR160A* and *PRE-MIR160C*, showing 3.0- and 2.0-fold upregulation, respectively, when compared to the level of expression of these two miR160 precursor transcripts in Col-0 plants (Figure 1E,G). In contrast, the abundance of the *PRE-MIR160B* was reduced by 1.8-fold in *drb2* roots compared to Col-0 (Figure 1F). Using stem-loop RT-qPCR, SL-RT-qPCR, miR160 abundance was revealed to be reduced by 5.0-fold in *drb1* roots (Figure 1H), which is likely the direct result of defective precursor transcript processing in the absence of DRB1 function (Figure 1E–G). The loss of DRB2 function in *drb2* roots had a mild effect on miR160 accumulation, and unlike *drb1*, miR160 abundance was increased by 1.5-fold compared to Col-0 roots (Figure 1H). This moderate elevation in miR160 accumulation in *drb2* roots is likely accounted for by the enhanced processing of the *PRE-MIR160B* precursor in the absence of DRB2 antagonism of the DRB1/DCL1 functional partnership.

The large change in miR160 precursor transcript abundance compared to the relatively mild alteration to mature miR160 abundance (Figure 1E–H) raised the possibility that other regulatory factors are involved in regulating the miR160 expression module. Previous studies have implicated endogenous target mimics (*eTM160-1* and *eTM160-2*) in adding to the regulatory complexity of the miR160 expression module in *Arabidopsis* [56]. These two *eTM*s are expressed in a spatiotemporal manner to finetune miR160 abundance [56]. We therefore assessed the expression of *eTM160-1* and *eTM160-2* in the roots of 3-week-old *drb1*, *drb2* and Col-0 plants. Compared to their abundance in Col-0 roots, the expression of *eTM160-1* and *eTM160-2* was reduced by 1.8- and 1.5-fold, respectively, in *drb1* roots (Figure 1I,J). A shared trend in the abundance of miR160 and its *eTM*s suggests that *eTM* transcript abundance was ‘*scaling*’ with that of its regulated miRNA, miR160 (Figure 1H–J). In *drb2* roots, the expression of *eTM160-1* was elevated 2.0-fold (Figure 1I), while the abundance of the *eTM160-2* transcript was mildly reduced by 1.4-fold (Figure 1J). The elevated abundance of both miR160 and *eTM160-1* in *drb2* roots again suggested that the endogenous target mimic scaled in abundance with its regulated miRNA, miR160.

In *drb1* roots, the 5.0-fold reduction in miR160 abundance correlated with increased expression of the *ARF10* (2.3-fold), *ARF16* (1.5-fold) and *ARF17* (2.5-fold) target genes (Figure 1K–M). In *drb2* roots, the level of expression of *ARF10*, *ARF16* and *ARF17* was elevated by 1.3-, 2.1- and 1.1-fold, respectively (Figure 1K–M), in response to the 1.5-fold increase in miR160 abundance (Figure 1H). Taken together, the molecular analyses presented in Figure 1 indicated that both the canonical DRB1-dependent and non-canonical DRB2-dependent miR160-directed mode of target gene expression regulation was required for the absolute control of *ARF10*, *ARF16* and *ARF17* transcript abundance in *Arabidopsis* roots. These results also indicated that DRB1 is the primary DRB partner of DCL1 for miR160 precursor transcript processing, playing a principal role in the regulation of *ARF10*, *ARF16* and *ARF17* expression in the roots of 3-week-old *Arabidopsis* plants (Figure 1K–M). In addition, the altered abundance of both the precursor and mature forms of miR160 in *drb2* roots suggested that DRB2 played a distinct role in the processing of miR160 from its precursors, which would result in potential additional layers of regulation complexity, including a translational repression mode of RNA silencing, operating in *Arabidopsis* roots to control the expression of *ARF10* and *ARF16*.

### 3.2. Phenotypic and Molecular Characterization of the Root Systems of Col-0 Plants and the Col-0/MIR160B, Col-0/mARF10 and Col-0/mARF16 Transformant Lines

To further understand the role of DRB2 in adding to the regulatory complexity of the miR160 expression module, three plant expression vectors were constructed and introduced into Col-0, *drb1* and *drb2* plants. These constructs included the (1) *PRE-MIR160B* overexpression vector (*MIR160B*), (2) miR160-resistant transgene version of *ARF10* (*mARF10*), and (3) miR160-resistant transgene version of *ARF16* (*mARF16*). Typical root phenotypes of the three transformant populations, and unmodified Col-0 plants are shown in Figure 2A–D. The average length of the primary root of Col-0/*MIR160B* transformants was 84 mm, representing a 121% increase to the primary root length of unmodified Col-0 plants (38 mm) (Figure 2B). A similar degree of promotion to primary root length was also observed for the Col-0/*mARF10* and Col-0/*mARF16* transformant lines, with increases of 123% to 85 mm, and 111% to 80 mm documented for these two transformant lines, respectively (Figure 2B). The introduction of the *MIR160B* transgene into Col-0 plants also significantly promoted lateral and adventitious root development, with Col-0/*MIR160B* plants developing an average of 34 lateral and 4 adventitious roots at 3 weeks of age compared to 12 and 1.5 lateral and adventitious roots in unmodified Col-0 plants, which represented increases of 183% and 167%, respectively (Figure 2C,D). The Col-0/*mARF10* and Col-0/*mARF16* transformant lines displayed much milder changes in lateral and adventitious root development than did the Col-0/*MIR160B* transformant line (Figure 2C,D). That is, 3-week-old Col-0/*mARF10* plants developed an average of 15 lateral roots and 2.1 adventitious roots, representing increases of 25% and 40% compared to untransformed Col-0 plants. Similarly, 3-week-old Col-0/*mARF16* transformants showed mild increases in lateral root (15.5 lateral roots or a 29% increase) and adventitious root (2.8 adventitious roots or a 87% increase) development in comparison to unmodified Col-0 plants of the same age (Figure 2C,D).

Consistent with the 17.2-fold increase in miR160 abundance in Col-0/*MIR160B* roots, the expression of the two target genes, *ARF10* and *ARF16*, was repressed by 2.5- and 3.3-fold, respectively (Figure 2I,J). This moderate decrease in target gene expression compared to the strong 17.2-fold increase in miR160 abundance in Col-0/*MIR160B* roots could be accounted for by the increased abundance of the target mimics *eTM160-1* and *eTM160-2* which were potentially ‘*buffering*’ miR160-directed repression of target-gene expression. The Col-0/*mARF10* transformant showed a 4.3-fold increase in *ARF10* transcript abundance in its root tissues (Figure 2I), with *ARF16* expression being largely unaffected (Figure 2J). Similarly, in Col-0/*mARF16* roots, the level of *ARF16* transcript was elevated by 4.7-fold with *ARF10* expression levels being similar to Col-0 roots (Figure 2I,J). Although miR160 levels were revealed by SL-RT-qPCR to be elevated by 2.8- and 1.8-fold in Col-0/*mARF10* and Col-0/*mARF16* roots, respectively (Figure 2F), the highly increased abundance of the two miR160 target transcripts in the respective transformant lines was consistent with the modified transcripts being resistant to miR160 binding, and therefore, miR160-directed expression repression. This molecular alteration to the miR160 expression module also likely accounted for the phenotypic changes in the root systems of these transformant lines.

It is important to note here that in the roots of the Col-0/*mARF10* transformant line, the 2.8-fold elevation in miR160 abundance correlated with a 2.0-fold reduction in *ARF17* expression (Appendix A), indicating that the miR160-directed target transcript cleavage mode of RNA silencing of the *ARF17* transcript was enhanced. Interestingly, however, in Col-0/*mARF16* roots, the 1.8-fold increase in miR160 abundance (Figure 2E) failed to alter the expression of the *ARF17* target gene (Appendix A). In addition, in all three transformant populations, the expression of *DRB1*, *DRB2* or *DRB4* was not altered in the roots compared to their respective levels of expression in unmodified Col-0 plants (Appendix A). When considered together, RT-qPCR analysis suggested that alteration to the abundance of either the targeting miRNA (miR160), or its target genes (*ARF10* and *ARF16*), failed to induce a feedback mechanism to promote *DRB1*, *DRB2* or *DRB4* gene expression in Col-0 roots to compensate for the molecular alterations introduced into the miR160 expression module in Col-0 plants.

### 3.3. Phenotypic and Molecular Characterization of the Root System of the drb1 Single Mutant and Transformant Lines drb1/MIR160B, drb1/mARF10 and drb1/mARF16

Compared to the unmodified *drb1* mutant, the introduction of the *MIR160B*, *mARF10* and *mARF16* transgenes caused considerable alteration of the root architecture (Figure 3A). More specifically, the average primary root length of the *drb1*/*MIR160B* transformant was increased by 44% to 36 mm from 25 mm for *drb1* plants (Figure 3B), and the average number of lateral roots was increased by 200% to 12 lateral roots from 4 lateral roots for *drb1* plants (Figure 3C). However, the number of adventitious roots per *drb1*/*MIR160B* plant remained unchanged from that of *drb1* plants (Figure 3D). The *drb1*/*mARF10* transformant showed enhanced root development in all three assessed root phenotypic metrics when compared to 3-week-old *drb1* plants (Figure 3A–D), with average primary root length of 33 mm (33% increase), an average of 10 lateral roots (150% increase) and 6.7 adventitious roots (67.5% increase). Similarly, the development of all three assessed root phenotypic metrics was promoted in 3-week-old *drb1*/*mARF16* plants with 33 mm average primary root length (33% increase), 9.5 lateral roots (138% increase) and 5.4 adventitious roots (35% increase) per *drb1*/*mARF16* plant (Figure 3B–D).

In *drb1*/*MIR160B* roots, the overexpression of *PRE-MIR160B* resulted in a 79-fold increase in miR160 precursor transcript levels (Figure 3E). However, surprisingly, the abundance of mature miR160 remained unchanged in *drb1*/*MIR160B* roots (Figure 3F). This result indicates that DRB1 function is absolutely required for efficient processing of the *PRE-MIR160B* precursor transcript by DCL1 in the roots of 3-week-old *Arabidopsis* plants. The transcript abundance of the miR160 target mimics *eTM160-1* was significantly elevated by 10.4-fold, whereas that of *eTM160-2* was reduced by 2.0 fold, in the roots of *drb1*/*MIR160B* plants (Figure 3G,H). The strongly increased abundance of *eTM160-1* could partly account for the unchanged miR160 level in *drb1*/*MIR160B* plants due to increased sequestering or buffering of the miRNA. Consistent with the stable miR160 level, the expression of the two *ARF* genes assessed, *ARF10* and *ARF16*, remained unchanged in the roots of the *drb1*/*MIR160B* transformant line compared to unmodified *drb1* plants (Figure 3I,J).

In the roots of the *drb1*/*mARF10* transformant, the level of both *PRE-MIR160B* and its processed sRNA, miR160, was significantly elevated by 3.2- and 4.0-fold, respectively, compared to the *drb1* mutant (Figure 3E,F). The level of *eTM160-1* expression was also significantly elevated by 5.9-fold in *drb1*/*mARF10* roots (Figure 3G), whereas the *eTM160-2* transcript level remained largely unchanged (up by 1.1-fold) (Figure 3H). As expected, a significant 2.4-fold increase in *ARF10* expression was detected in *drb1*/*mARF10* roots (Figure 3I). Interestingly, a mild 1.8-fold increase in *ARF16* expression was observed in the root system of *drb1*/*mARF10* plants. Like the *drb1*/*mARF10* plants, the expression of the *mARF16* transgene in the roots of 3-week-old *drb1*/*mARF16* transformant induced a 4.0-fold increase in *PRE-MIR160B* expression (Figure 3E), which in turn led to the 4.4-fold increase in miR160 abundance compared to untransformed *drb1* plants (Figure 3F). Also similar to the *drb1*/*mARF10* plants, increased miR160 abundance in *drb1*/*mARF16* roots was associated with a 5.7-fold increase in *eTM160-1* transcript abundance, and a 1.4-fold decrease in *eTM160-2* abundance (Figure 3G,H). In addition, the over-expression of the *mARF16* transgene in *drb1*/*mARF16* roots led to a significant 3.2-fold increase in *ARF16* expression and a moderate 1.6-fold elevation in *ARF10* expression (Figure 3I,J). Taken together, the expression profiles of the individual components of the miR160 expression module in the roots of the *drb1*/*mARF10* and *drb1*/*mARF16* transformants suggested that over-expression of the miR160-resistant *ARF* genes induced a positive feedback response at all levels of the targeted miRNA expression module to accommodate the introduced molecular modifications in these plant lines.

In contrast to the considerable alterations to root architecture, the *drb1*/*MIR160B*, *drb1*/*mARF10* and *drb1*/*mARF16* transformants only displayed mild changes in aerial tissue development compared to *drb1* plants. Specifically, rosette leaf area was increased by 19% to 1.9 cm^2^ in *drb1*/*MIR160B* plants, moderately decreased by 31% to 1.1 cm^2^ in *drb1*/*mARF10* plants, and remained unchanged at 1.6 cm^2^ in *drb1*/*mARF16* plants (Appendix A). Notably, all three miR160 precursor transcripts (*PRE-MIR160A*, *PRE-MIR160B* and *PRE-MIR160C*) showed increased accumulation in the roots of *drb1*/*MIR160B*, *drb1*/*mARF10* and *drb1*/*mARF16* transformants when compared to the 3-week-old *drb1* plants (Appendix A). However, the level of mature miR160 was unchanged in the *drb1*/*MIR160B* roots, whereas the *drb1*/*mARF10* and *drb1*/*mARF16* plants showed 4.0- and 4.4-fold increase in miR160 abundance, respectively, compared to the untransformed *drb1* plant (Figure 3F). Consistent with the unchanged miR160 abundance, the expression level of *ARF17* remained unchanged in *drb1*/*MIR160B* roots (Appendix A). Surprisingly, the level of *ARF17* expression also remained largely unchanged in the roots of the *drb1*/*mARF10* and *drb1*/*mARF16* plants despite the greater than 4-fold increase in miR160 levels in these two transformant lines. This result raised the possibility that DRB1 activity is required for efficient miR160-directed *ARF17* target transcript cleavage, and in the absence of DRB1 function, enhanced *ARF17* transcript cleavage due to increased miR160 abundance could not be achieved. This finding also suggests that in *Arabidopsis* roots, *ARF17* transcript abundance is solely regulated by a DRB1-dependent miR160-directed transcript cleavage mode of RNA silencing.

*DRB2* and *DRB4* expression was only mildly affected in *drb1*/*mARF10* and *drb1*/*mARF16* roots, with respective 1.2- and 1.4-fold increases for *DRB2* expression (Appendix A) and 1.1- and 1.4-fold increases for *DRB4* expression (Appendix A). RNA secondary structure prediction showed that the *PRE-MIR160A* and *PRE-MIR160C* stem-loop structures possess a highly compacted loop, with only 3 and 4 unpaired nucleotides, respectively (Appendix A). By comparison, the loop structure in the *PRE-MIR160B* transcript is less compact, composed of 7 unpaired nucleotides. The relatively compact loop region structure of *PRE-MIR160A* and *PRE-MIR160C* could have potentially enabled more efficient processing of these two precursors by DRB2/DCL1 or DRB2/DCL4 partnership in the absence of a canonical DRB1/DCL1 partnership [7,9,29,30], which in turn could account for the 4.0- and 4.4-fold increase to miR160 abundance in the *drb1*/*mARF10* and *drb1*/*mARF16* transformants (Figure 3F).

### 3.4. Phenotypic and Molecular Characterization of the Root System of the drb2 Single Mutant and the drb2/MIR160B, drb2/mARF10 and drb2/mARF16 Transformant Lines

The expression changes of components of the miR160 expression module, including *PRE-MIR160B*, miR160, *ARF10* and *ARF16* in the roots of *drb2* plants (Figure 1), prompted us to investigate the consequence of introducing the *MIR160B*, *mARF10* and *mARF16* transgenes into the *drb2* background. As shown in Figure 4A, the roots of the *drb2*/*MIR160B* transformant line showed no significant change in overall root architecture when compared to the untransformed *drb2* mutant. In contrast, the *drb2*/*mARF10* transformant displayed dramatically altered root development (Figure 4A), showing primary root length of 97 mm (19.8% increase) (Figure 4B), and an average of 38.5 lateral (131% increase) and 3.9 adventitious (69.6% increase) roots (Figure 4C,D). All three quantified root phenotype metrics were also altered in the *drb2*/*mARF16* transformant line, however, the degree of alteration was mild compared to *drb2*/*mARF10* plants, with an average primary root length of 87 mm (7.4% increase), an average lateral root number of 28.5 (35.7% increase), and adventitious root number decreased by 17.4% from 2.3 in *drb2* plants to 1.9 in *drb2*/*mARF16* plants (Figure 4B–D).

Molecular profiling of *drb2*/*MIR160B* roots revealed that *PRE-MIR160B* transcript abundance was reduced by 1.8-fold, and that the accumulation of the miR160 sRNA was significantly elevated by 17.0-fold in comparison to the untransformed *drb2* plants (Figure 4E,F). The expression level of *eTM160-1* and *eTM160-2* was moderately, yet significantly reduced by 2.2- and 1.9-fold, respectively (Figure 4G,H). This formed an unexpected result as elevated miR160 abundance in *drb2*/*MIR160B* roots was expected to lead to elevated eTM transcript abundance, and not decreased eTM abundance as determined by RT-qPCR., The 17.0-fold increase in miR160 abundance also correlated with enhanced repression of *ARF10* and *ARF16* expression, with the abundance of these two miR160 target transcripts reduced by 2.0- and 4.0-fold in the roots of *drb2*/*MIR160B* plants, respectively (Figure 4I,J). In summary, the data presented in Figure 4E–J shows that *PRE-MIR160B* transcript abundance was decreased in the *drb2*/*MIR160B* root system (Figure 4E), indicating that in the absence of DRB2 antagonism of DRB1 function, *PRE-MIR160B* is more efficiently processed by the DRB1/DCL1 functional partnership. This in turn led to elevated miR160 accumulation, and therefore, enhanced degrees of miR160-directed repression of its target genes, *ARF10* and *ARF16* (Figure 4G–J). Furthermore, the expression data presented in Figure 4E–J clearly reveals that DRB2-mediated mechanisms add an additional layer of regulatory complexity as part of the absolute control of *ARF10* and *ARF16* expression by miR160 in *Arabidopsis* roots.

In the roots of the *drb2*/*mARF10* transformant, *PRE-MIR160B* transcript abundance was only marginally elevated by 1.1-fold (Figure 4E), which was associated with a 2.6-fold increase in miR160 abundance compared to the roots of 3-week-old *drb2* plants (Figure 4F). The increased miR160 abundance caused a 2.0-fold reduction in *eTM160-1* expression (Figure 4G). Surprisingly, the level of *eTM160-2* expression was increased by 1.9-fold which was likely to be in response to the increased miR160 accumulation in *drb2*/*mARF10* roots (Figure 4H). As expected, the expression of *ARF10* was elevated in *drb2*/*mARF10* roots by 3.4-fold (Figure 4I). In contrast to elevated *ARF10* abundance, the expression level of *ARF16* was reduced by 2.7-fold (Figure 4J), which was likely due to the higher level of miR160 directing increased repression of the expression of its target gene. In *drb2*/*mARF16* roots, *PRE-MIR160B* expression was moderately reduced by 1.8-fold, and the abundance of miR160 was significantly increased by 4.6-fold, compared to *drb2* roots (Figure 4E,F). This result suggested that the *PRE-MIR160B* precursor transcript is more readily processed in *drb2*/*mARF16* than *drb2* root tissues, leading to the enhanced accumulation of the miR160 sRNA. Consistent with the elevated miR160 abundance, the expression of *eTM160-1* was significantly reduced by 5.0-fold (Figure 4G), and *eTM160-2* transcript abundance slightly decreased by 1.1-fold (Figure 4H) in *drb2*/*ARF16* roots. *ARF10* expression was also decreased by 4.0-fold (Figure 4I), and *ARF16* transcript abundance was elevated by 2.7-fold in the root system of the *drb2*/*mARF16* transformant line.

Compared to unmodified 3-week-old *drb2* plants, the introduction of all three transgenes repressed aerial tissue growth (Appendix A). Total rosette leaf area was decreased by 43%, 12% and 51% to 2.3 cm^2^, 3.6 cm^2^ and 2.0 cm^2^ in the *drb2*/*MIR160B*, *drb2*/*mARF10* and *drb2*/*mARF16* transformant lines, respectively, compared to 4.1 cm^2^ for the unmodified *drb2* mutant (Appendix A). The abundance of the *PRE-MIR160A* precursor transcript was reduced by 2.3-, 2.9- and 2.5-fold in the roots of the *drb2*/*MIR160B*, *drb2*/*mARF10* and *drb2*/*mARF16* transformant lines (Appendix A), which correlated with significantly increased miR160 accumulation (Figure 4F), suggesting an increased efficiency of miR160 precursor processing in the roots of these transformants. Interestingly, the abundance of *PRE-MIR160C* transcript was similar in the roots of the *drb2*/*MIR160B*, *drb2*/*mARF10* and *drb2*/*mARF16* transformant lines compared to the unmodified *drb2* mutant (Appendix A). This suggested that DRB2 is not involved in regulating the biogenesis of miR160 from the *PRE-MIR160C* precursor transcript in the roots of 3-week-old *Arabidopsis* plants. Consistent with the increased miR160 abundance, *ARF17* expression was considerably reduced by 2.1-, 3.3- and 2.9-fold, respectively, in *drb2*/*MIR160B*, *drb2*/*mARF10* and *drb2*/*mARF16* roots (Appendix A). In addition, *DRB1* transcript abundance was increased by 3.2-, 1.8- and 1.9-fold, respectively, in these three transformant lines (Appendix A). The highest degree of promotion of *DRB1* expression in the roots of the *drb2*/*MIR160B* transformant line once again suggested that in the absence of DRB2 antagonism, the ability of DRB1 to form a functional partnership with DCL1 for efficient precursor transcript processing is enhanced. Interestingly, *DRB4* expression was increased by a similar degree (approximately 1.5-fold) in the roots of the *drb2*/*MIR160B*, *drb2*/*mARF10* and *drb2*/*mARF16* transformant lines (Appendix A). Elevated *DRB4* expression in all three transformant lines of the *drb2* mutant background again identifies the importance of DRB2 antagonism on DRB4 function to ensure the correct level of regulatory complexity for the miR160 expression module in the root system of 3-week-old *Arabidopsis* plants.

## 4. Discussion

### 4.1. Interplay between DRB1 and DRB2 Is Required for the Appropriate Regulatory Control of the miR160 Expression Module in Arabidopsis Roots

Of the five members of the *Arabidopsis DRB* gene family, DRB1 is firmly established as the primary functional partner of DCL1, the miRNA pathway-specific DCL endonuclease [6,8,9]. In *Arabidopsis*, as with all plant species, miRNAs are master regulators of gene expression throughout development, including coordinating the transition from juvenile to adult vegetative development, and the subsequent transition from vegetative growth to reproductive development [57,58,59]. The retarded development of the *drb1* root system displayed by 3-week-old *drb1* plants (Figure 1) therefore formed an expected result, and confirmed the abnormalities associated with all aspects of *Arabidopsis* development in the absence of DRB1 function [6,26,28]. At three weeks of age, defective root development of the *drb1* single mutant was characterized by reductions to primary root length and lateral root number in combination with promotion of adventitious root development (Figure 1B–D). Considering that molecular alteration to the miR160 expression module, including altered abundance of miR160 or of its *ARF10*, *ARF16* and *ARF17* target genes has been previously associated with changes in *Arabidopsis* root development [49,50,55], we applied here a standard RT-qPCR approach to molecularly assess this expression module. Elevated abundance of *PRE-MIR160A*, *PRE-MIR160B* and *PRE-MIR160C*, together with reduced miR160 accumulation in the roots of 3-week-old *drb1* plants (Figure 1E–H), readily confirmed the absolute requirement of DRB1 function for efficient miRNA precursor processing by DCL1 for miR160 production [6,8,9,20,21,22,23,26]. Furthermore, an elevated degree of *ARF10*, *ARF16* and *ARF17* expression (Figure 1K–M) in response to the reduced miR160 abundance (Figure 1H), showed that the canonical DRB1-dependent mRNA cleavage mode of RNA silencing [12,13,26,34] formed the primary mechanism of miR160-directed gene expression regulation in the root system of 3-week-old wild-type *Arabidopsis* plants.

Previous research has shown that developmental defects in the *drb2* mutant are relatively mild in comparison to those displayed by the *drb1* mutant [28,30,31]. However, the data presented in Figure 1 provides further solid evidence of the requirement of DRB2 function for normal *Arabidopsis* development. Specifically, primary root elongation, and the formation of lateral and adventitious roots were all promoted in 3-week-old *drb2* plants. Our previous research has repeatedly demonstrated the requirement of DRB2 for producing specific miRNA cohorts in the tissues where *DRB2* is expressed in Col-0 plants [30,31,33,34]. Here we provide further evidence to indicate that DRB2 functions both synergistically and antagonistically on the DRB1/DCL1 functional partnership for miR160 production. More specifically, the increased abundance of *PRE-MIR160A* and *PRE-MIR160C* in *drb2* roots (Figure 1E,G) indicates that these miR160 precursor transcripts are processed at a reduced level of efficiency by DCL1 in the absence of DRB2 activity. This demonstrates that DRB2 acts in a synergistic manner with DRB1 to produce miR160 from these two precursor transcripts in *Arabidopsis* roots. In contrast, the reduced level of *PRE-MIR160B* abundance in *drb2* roots (Figure 1F) suggests that DRB2 acts antagonistically on the DRB1/DCL1 partnership for miR160 processing from this precursor transcript, that is; in the absence of competitive binding of *PRE-MIR160B* by DRB2, the *PRE-MIR160B* precursor is more readily available for processing by the DRB1/DCL1 functional partnership.

Pélissier et al. [29] have shown that in addition to DRB4, together with its functional partner protein DCL4, DRB2 is required for the production of the newly-evolved class of *Arabidopsis* miRNAs from their highly complementary stem-loop structured precursors. Comparison of the folding structures of the three miR160 precursors (Appendix A) shows that *PRE-MIR160A* and *PRE-MIR160C* form stem-loop structures that more closely resemble those adopted by precursors of the newly-evolved miRNAs [32] than does the *PRE-MIR160B* precursor. This suggests that the folding structure formed by an individual miRNA precursor transcript likely forms the primary determinant of the degree of involvement, and therefore, the mechanism of action by DRB2 (i.e., DRB2 antagonism or synergism), to fine tune the rate of production of a DRB2-dependent miRNA in *Arabidopsis*. RT-qPCR also showed both elevated abundance of miR160 and increased expression of *ARF10* and *ARF16* in the roots of 3-week-old *drb2* plants (Figure 1). This shared trend in both miR160 abundance and its target gene expression raised the possibility that translational repression is operating as the mode in miR160-mediated gene repression [27,31,33,34]. Thus, in *Arabidopsis* roots, miR160 may direct both the canonical mode of RNA silencing, mRNA cleavage, and the noncanonical mode of RNA silencing, translational repression, to regulate *ARF10* and *ARF16* expression in this tissue.

### 4.2. Development of the Arabidopsis Root System Is Altered in Plant Lines That Harbor Molecular Modifications to the miR160 Expression Module

The initial phenotypic and molecular analyses performed on the roots of 3-week-old Col-0, *drb1* and *drb2* plants strongly inferred that the involvement of DRB2 on the miR160 expression module was limited to specific components of the module, including regulation of the rate of miR160 production from the *PRE-MIR160B* precursor transcript, and by potentially mediating translational repression as a mode of miR160-directed RNA silencing to add an additional layer of regulation to afford a tighter degree of control over the level of *ARF10* and *ARF16* gene expression. To gain further insight into the role(s) of DRB2 in the miR160 expression module as part of root development, the *MIR160B*, *mARF10* and *mARF16* transgenes were constructed and introduced in Col-0, *drb1* and *drb2* plants.

Compared to unmodified 3-week-old Col-0 plants, of the three root system metrics assessed in the Col-0/*MIR160*, Col-0/*mARF10* and Col-0/*mARF16* transformant lines (Figure 2B–D), altered primary root morphology formed the most pronounced phenotypic change (Figure 2B). The similar phenotypic alterations between the miR160 overexpression and miRNA-resistant target gene transformants was surprising, as decoupling *ARF10* and *ARF16* posttranscriptional regulation by miR160 in Col-0/*mARF10* and Col-0/*mARF16* plants was expected to have the opposite phenotypic effect to the enhancement of miR160-directed *ARF* gene expression repression in Col-0/*MIR160B* plants. For example, at the molecular level it was also surprising that elevated miR160 abundance in Col-0/*mARF10* and Col-0/*mARF16* roots failed to influence the expression of the unmodified *ARF* target gene (Figure 2F,I,J). That is, although miR160 abundance was significantly elevated by 2.8-fold in Col-0/*mARF10* roots, *ARF16* expression remained largely unchanged (Figure 2F,I), and in Col-0/*mARF16* roots, *ARF10* expression remained at a similar level to that observed in unmodified Col-0 roots, even though miR160 abundance was elevated by 1.8-fold (Figure 2F,J). However, in contrast to *ARF10* and *ARF16* transcript abundance in these transformant lines, the extent to which the expression of the *ARF17* was decreased in Col-0/*mARF10*, Col-0/*mARF16* and Col-0/*MIR160B* roots (Appendix A) did correlate with elevated miR160 abundance (Figure 2F). Via a similar approach to that adopted here, Mallory and colleagues [60] showed that in *Arabidopsis* transformants which highly expressed the introduced *mARF17* transgene (i.e., transformant lines with a high abundance of the miR160-resistant *ARF17* transcript), primary root length was reduced. This previous report [60] provides strong support that enhancement of miR160-directed gene expression repression of *ARF17* was the likely cause of the promotion of primary root development in the Col-0/*MIR160B*, Col-0/*mARF10* and Col-0/*mARF16* transformant lines (Figure 2A,B and Appendix A).

Among the transformant lines in the Col-0 background, promotion to lateral root development was observed only in Col-0/*MIR160B* plants (Figure 2C). This phenotypic change appeared to be directly related to highly elevated miR160 abundance (Figure 2F) and reduced *ARF10*, *ARF16* and *ARF17* expression (Figure 2I,J and Appendix A). ARF17 is a well-known negative regulator of lateral root development [57,60,61], and therefore, enhanced miR160-directed repression of *ARF17* expression was the likely initiator of promoted lateral root development in Col-0/*MIR160B* plants. All three transformant lines generated in the *drb1* background developed an increased number of lateral roots (Figure 3C). This finding is significant considering that the unmodified *drb1* mutant has impeded lateral root development compared to Col-0 plants (Figure 1). While ARF10 and ARF16 play opposing roles in lateral root initiation [57], both ARFs promote the growth of lateral roots once lateral root initials have emerged through the root epidermis [62]. It is therefore likely that the increased expression of *ARF10* and *ARF16* in the *drb1*/*mARF10* and *drb1*/*mARF16* transformant lines (Figure 3I,J) can rescue the deleterious effects of disrupted DRB1 function on miR160-controlled lateral root development. Lateral root number was also increased in the *drb2*/*mARF10* and *drb2*/*mARF16* transformant lines (Figure 4C). In these two plant lines, *ARF16* and *ARF10* expression was reduced (Figure 4I,J) as a result of elevated miR160 abundance (Figure 4F) and therefore, enhancement of the miR160-directed target transcript cleavage (Appendix A). These results suggest that ARF10 and ARF16 can promote lateral root development independently of each other.

As observed with lateral root development for the transformant lines in the Col-0 background, promotion of adventitious root development appeared to be related to elevated miR160 accumulation (Figure 2D). Among the Col-0/*MIR160B*, Col-0/*mARF10* and Col-0/*mARF16* transformant lines, adventitious root development was promoted to the greatest extent in Col-0/*MIR160B* plants where miR160 accumulation increased to its highest level (Figure 2F). Accordingly, the expression of *ARF10*, *ARF16* and *ARF17* showed the most pronounced reduction in Col-0/*MIR160B* roots (Figure 2I,J and Appendix A). In adventitious roots, a previous report has shown that *ARF6* and *ARF8* expression is negatively regulated by ARF17 activity [63], thereby identifying ARF17 as a repressor of adventitious root development in *Arabidopsis* [60,61,63]. Therefore, reduced *ARF17* expression in Col-0/*MIR160B*, Col-0/*mARF10* and Col-0/*mARF16* roots due to increased miR160-directed repression likely accounts for the observed promotion of adventitious root development in these three transformant lines. Adventitious root development was also promoted in the *drb1*/*mARF10* and *drb1*/*mARF16* transformant lines (Figure 3D). Interestingly, in the roots of all three transformant lines generated in the *drb1* background, *ARF17* expression remained largely unchanged from its level in *drb1* roots (Appendix A). However, *ARF10* and *ARF16* expression was increased in both transformant lines which could indirectly promote adventitious root development via their reported transcriptional regulation of *ARF6* and *ARF8* gene expression, with both ARF6 and ARF8 previously identified as positive regulators of adventitious root development [63]. Altered adventitious root development in the transformant lines in the *drb2* mutant background was primarily restricted to the *drb2*/*mARF10* plant line (Figure 4D). Compared to *drb2* plants, miR160 abundance was significantly elevated in *drb2*/*mARF10* roots which readily accounted for the significantly reduced level of *ARF16* (Figure 4I) and *ARF17* expression (Appendix A) in the root system of this transformant line. Considering the known role for ARF16 and ARF17 [61,63] in negatively regulating the activity of ARF6 and/or ARF8, it is unsurprising that an increase in adventitious root formation was observed in *drb2*/*mARF10* plants.

### 4.3. The Interaction of miR160 with Its eTM Regulatory Transcript, eTM160-1, Appears to Be via Distinct Mechanisms in the drb1 and drb2 Mutant Backgrounds

The first functional *eTM* identified in plants was *INDUCED BY PHOSPHATE STARVATION1* (*IPS1*), a non-cleavable ncRNA which was shown to add to the regulatory complexity of the miR399/*PHOSPHATE2* expression module [64]. Since this initial discovery, numerous other *eTM*s have been identified in *Arabidopsis* [56], and in other plants species such as rice and upland cotton [56,65,66]. The *Arabidopsis* miR160 is one such miRNA for which an *eTM* has been identified and shown to add regulatory complexity to the miR160–*ARF* target gene interaction [56]. More specifically, the *Arabidopsis* miR160 expression module is under additional regulatory scrutiny by *eTM160-1* and *eTM160-2,* which sequester miR160 activity in a spatiotemporal manner [56]. The RT-qPCR analyses presented here show that in unmodified *drb1* roots, reduced miR160 abundance was associated with decreased expression of both miR160-specific *eTM*s (Figure 1H–J). Similarly, in *drb2* roots, an elevated level of miR160 was associated with increased abundance of the *eTM160-1* transcript (Figure 1H,I). However, *eTM160-2* expression was reduced in response to elevated miR160 abundance in *drb2* roots (Figure 1J). This initial profiling exercise of *eTM* expression in unmodified *drb1* and *drb2* plants, indicated that in the root system of 21-day-old *Arabidopsis* plants, the *eTM160-1* ncRNA occupied the role of the primary non-cleavable target decoy of miR160-directed *ARF* target gene expression regulation via scaling in its abundance in accordance with its regulated miRNA (Figure 1H,I).

In response to elevated miR160 abundance in Col-0/*MIR160B*, Col-0/*mARF10* and Col-0/*mARF16* roots (Figure 2F), the level of the *eTM160-1* transcript was again observed to scale closely with that of its regulated miRNA (Figure 2G). This shared expression trend again suggested that a feedback mechanism was operating in the roots of Col-0/*MIR160B*, Col-0/*mARF10* and Col-0/*mARF16* plants with elevated miR160 abundance promoting the transcriptional activity of the locus from which the *eTM160-1* is transcribed. In the roots of *drb1*/*mARF10* and *drb1*/*mARF16* transformants, *eTM160-1* levels were again observed to scale in accordance with miR160 abundance. The abundance of the *eTM160-1* ncRNA was also elevated in the roots of the *drb1*/*MIR160B* transformant line even though miR160 abundance was determined to remain unchanged from its levels in unmodified *drb1* plants (Figure 3F,G). However, in this transformant line, the abundance of the *PRE-MIR160B* precursor transcript was dramatically increased by 79-fold (Figure 3E) to potentially suggest that *eTM160-1* expression may be responsive to both a change in mature miR160 abundance, and the transcriptional activity of the *MIR* genes from which miR160 precursor transcripts are transcribed. Such multitiered responsive would ensure that the tight regulatory control over the miR160 expression module is maintained in the *Arabidopsis* root system. Interestingly, in contrast to unmodified Col-0, *drb1* and *drb2* roots, and the roots of the *MIR160B*, *mARF10* and *mARF16* transformant lines in the Col-0 and *drb1* genetic backgrounds, an opposing abundance trend for the miR160 and *eTM160-1* transcripts was documented for the three transformant lines generated in the *drb2* mutant background (Figure 4F,G). This finding indicates that the molecular alterations introduced into the miR160 expression module had a more pronounced negative impact on this regulatory transcript than did the identical molecular alterations following their introduction into either Col-0 plants or the *drb1* mutant. More specifically, the introduction of the *MIR160B*, *mARF10* and *mARF16* transgenes into the *drb2* mutant background and the *in planta* expression of these transgenes in a plant line where DRB2 function is absent, somehow rendered the regulatory capacity of the *eTM160-1* transcript to the miR160 sRNA defective. This finding indicates that a higher level of complexity may be required to appropriately control the involvement of DRB2 in the miR160 expression module in *Arabidopsis* roots.

## 5. Conclusions

In this study, we used a molecular modification approach to determine the degree of involvement of DRB2 in regulating the miR160 expression module with our analyses revealing that DRB2 adds an additional layer of regulatory complexity to miR160 production from its precursor transcripts. Namely, DRB2 acts synergistically to the DRB1/DCL1 partnership for miR160 processing from *PRE-MIR160A* and *PRE-MIR160C*, and is antagonistic to DRB1/DCL1 function for miR160 production from *PRE-MIR160B*. The molecular alterations introduced into the Col-0, *drb1* and *drb2* plant lines, altered the architecture of the *Arabidopsis* root system, with the *in planta* expression of the *MIR160B*, *mARF10* and *mARF16* transgenes promoting the elongation of the primary root and enhancing lateral and adventitious root formation. Molecular assessment of the individual components of the miR160 expression module in transformants Col-0/*MIR160B*, Col-0/*mARF10*, Col-0/*mARF16*, *drb1*/*MIR160B*, *drb1*/*mARF10*, *drb1*/*mARF16*, *drb2*/*MIR160B*, *drb2*/*mARF10* and *drb2*/*mARF16*, and comparison of the obtained molecular profiles to those constructed for the root systems of unmodified Col-0, *drb1* and *drb2* plants, indicated that promotion of primary root elongation was the result of enhanced miR160-directed expression repression of *ARF17* expression. Molecular profiling of the miR160 expression module in this transformant series also showed that promotion of lateral root development was likely the result of altered miR160-directed expression regulation of *ARF17*, and to a lesser degree, *ARF10* and *ARF16*. Similarly, enhanced miR160-directed expression regulation of *ARF10*, *ARF16* and/or *ARF17*, was the likely initiator of promoting adventitious root development in individual transformant lines expressing either the *MIR160B*, *mARF10* or *mARF17* transgene. Finally, we also show that the ability of the eTM, *eTM160-1*, to sequester the activity of miR160 in *Arabidopsis* roots is impaired when further molecular modifications to the miR160 expression module are introduced into the *drb2* mutant background: a finding that readily demonstrates the high degree of regulatory complexity involved in controlling the miR160 expression module in *Arabidopsis* roots.

## Figures and Tables

**Figure 1 genes-15-01042-f001:**
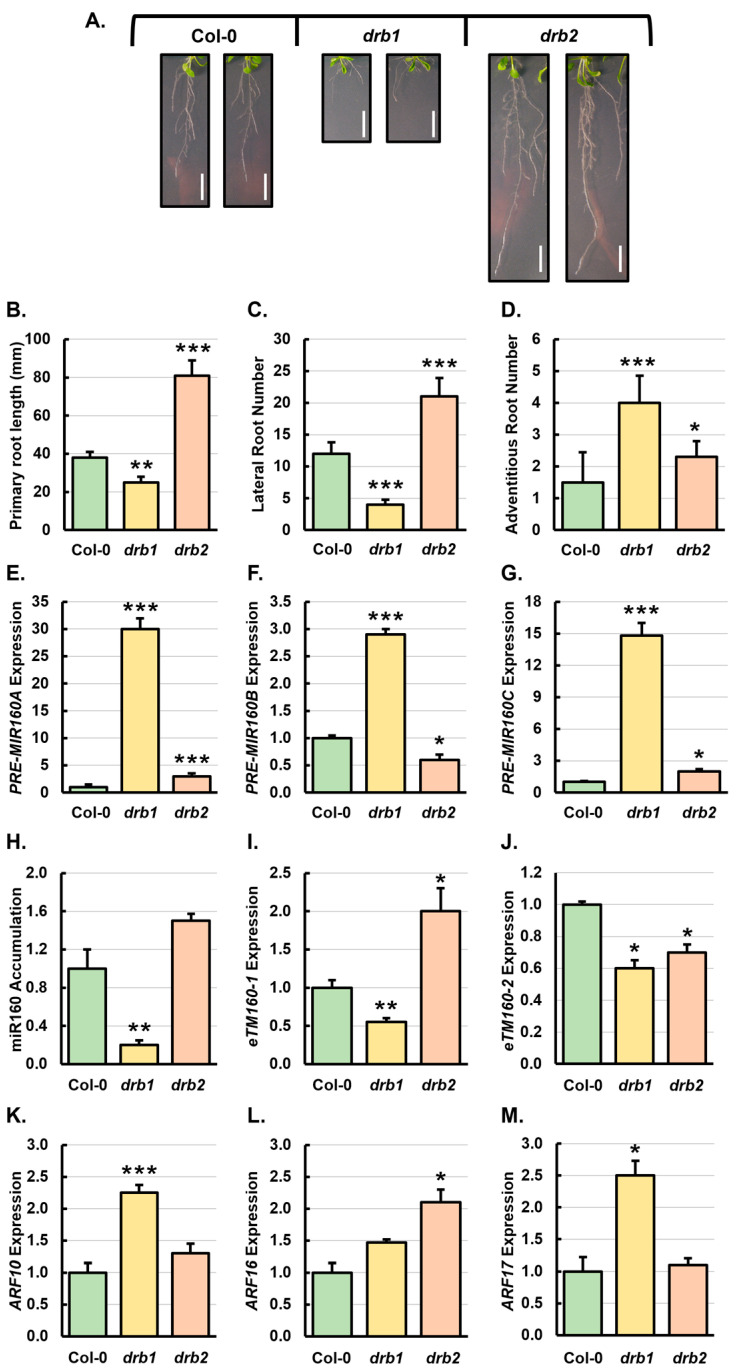
Phenotypic analysis of root system architecture of 3-week-old Col-0, *drb1* and *drb2* plants and molecular assessment of the miR160/*ARF10*/*ARF16*/*ARF17* expression module. (**A**) Root system development of 3-week-old Col-0, *drb1* and *drb2* plants. Scale bar = 1.0 cm. Quantification of primary root length (**B**), lateral root number (**C**) and adventitious root number (**D**) in 3-week-old Col-0, *drb1* and *drb2* plants. (**E**–**M**) RT-qPCR quantification of the level of expression of *PRE-MIR160A* (**E**), *PRE-MIR160B* (**F**), *PRE-MIR160C* (**G**), miR160 (**H**), *eTM160-1* (**I**), *eTM160-2* (**J**), *ARF10* (**K**), *ARF16* (**L**) and *ARF17* (**M**) in the roots of 3-week-old Col-0, *drb1* and *drb2* plants. (**E**–**M**) Fold changes were determined by the 2^−∆∆CT^ method with the use of three biological replicates which contained six pooled plants per replicate. Averages of expression are represented as a fold change for each assessed transcript and were compared to the values obtained for Col-0 plants by a standard two-tailed *t*-test. Error bars represent the standard error of the mean (SEM) and asterisks show * *p* ≤ 0.05, ** *p* ≤ 0.01, *** *p* ≤ 0.001.

**Figure 2 genes-15-01042-f002:**
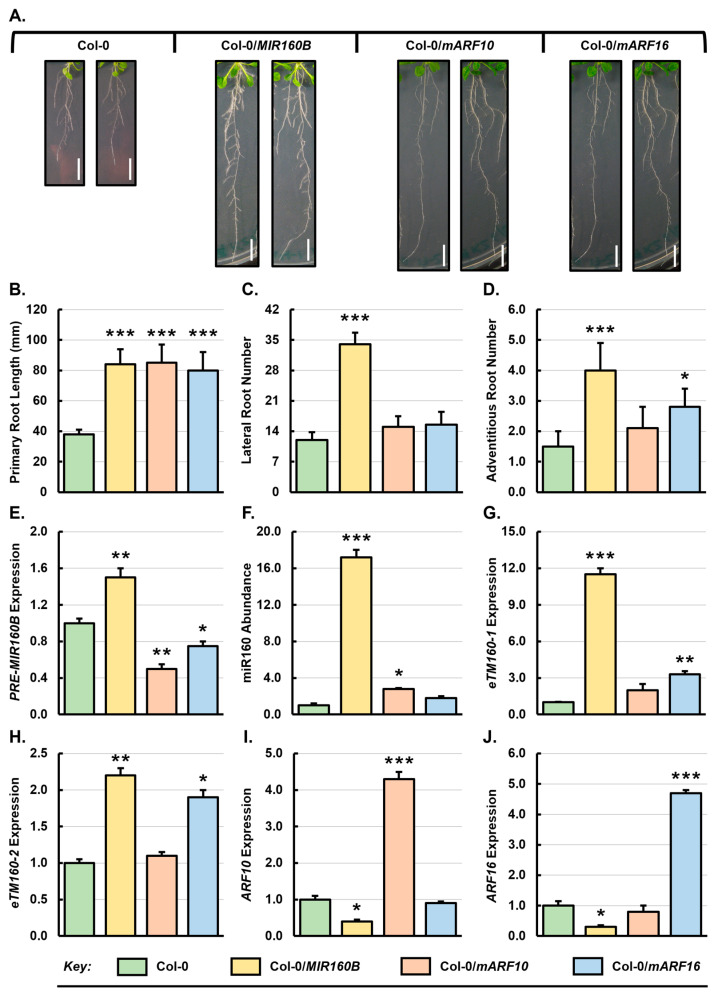
Phenotypic and molecular assessment of the miR160 expression module in Col-0 plants and the Col-0/*MIR160B*, Col-0/*mARF10*, and Col-0/*mARF16* transformant lines. (**A**) Typical root system architecture displayed by 3-week-old Col-0 plants and the Col-0/*MIR160B*, Col-0/*mARF10* and Col-0/*mARF16* transformant lines. Scale bar = 1.0 cm. Quantification of primary root length (**B**), lateral root number (**C**), and adventitious root number (**D**) in 3-week-old Col-0, Col-0/*MIR160B*, Col-0/*mARF10* and Col-0/*mARF16* plants. (**E**–**J**) RT-qPCR quantification of the level of expression of *PRE-MIR160B* (**E**), miR160 (**F**), *eTM160-1* (**G**), *eTM160-2* (**H**), *ARF10* (**I**), and *ARF16* (**J**) in the roots of 3-week-old Col-0, Col-0/*MIR160B*, Col-0/*mARF10* and Col-0/*mARF16* plants. (**E**–**J**) Fold changes were determined by the 2^−∆∆CT^ method with the use of three biological replicates which contained six pooled plants per replicate. Averages of expression are represented as a fold change for each assessed transcript and were compared to the values obtained for Col-0 plants by a standard two-tailed *t*-test. Error bars represent the standard error of the mean (SEM) and asterisks show * *p* ≤ 0.05, ** *p* ≤ 0.01, *** *p* ≤ 0.001.

**Figure 3 genes-15-01042-f003:**
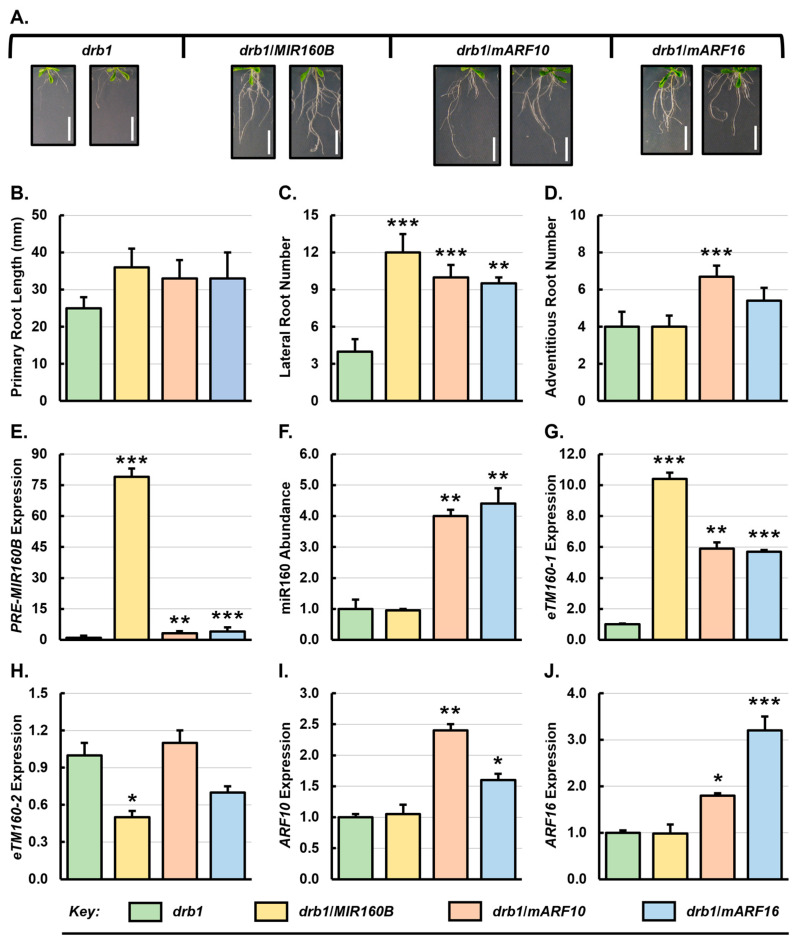
Phenotypic and molecular assessment of the miR160 expression module in the *drb1* single mutant and the *drb1*/*MIR160B*, *drb1*/*mARF10*, and *drb1*/*mARF16* transformant lines. (**A**) Typical root system architecture displayed by 3-week-old *drb1* plants and the *drb1*/*MIR160B*, *drb1*/*mARF10* and *drb1*/*mARF16* transformant lines. Scale bar = 1.0 cm. Quantification of primary root length (**B**), lateral root number (**C**), and adventitious root number (**D**) in 3-week-old *drb1*, *drb1*/*MIR160B*, *drb1*/*mARF10* and *drb1*/*mARF16* plants. RT-qPCR quantification of the level of expression of *PRE-MIR160B* (**E**), miR160 (**F**), *eTM160-1* (**G**), *eTM160-2* (**H**), *ARF10* (**I**), and *ARF16* (**J**) in the roots of 3-week-old *drb1*, *drb1*/*MIR160B*, *drb1*/*mARF10* and *drb1*/*mARF16* plants. (**E**–**J**) Fold changes were determined by the 2^−∆∆CT^ method with the use of three biological replicates which contained six pooled plants per replicate. Averages of expression are represented as a fold change for each assessed transcript and were compared to the values obtained for *drb1* plants by a standard two-tailed *t*-test. Error bars represent the standard error of the mean (SEM) and asterisks show * *p* ≤ 0.05, ** *p* ≤ 0.01, *** *p* ≤ 0.001.

**Figure 4 genes-15-01042-f004:**
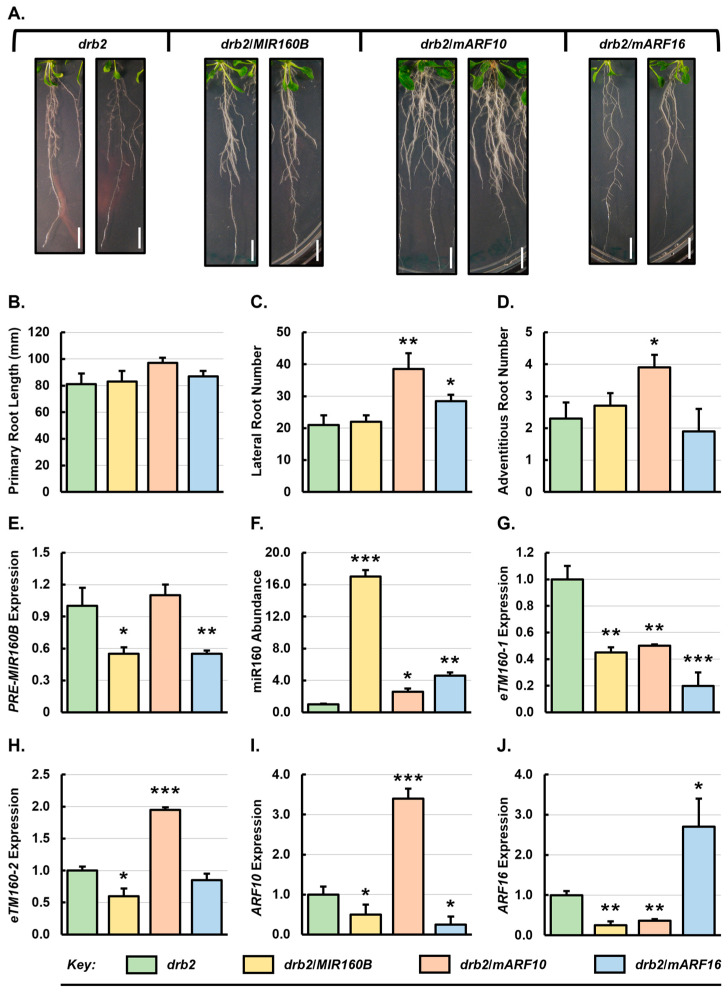
Phenotypic and molecular assessment of the miR160 expression module in the *drb2* single mutant and the *drb2*/*MIR160B*, *drb2*/*mARF10*, and *drb2*/*mARF16* transformant lines. (**A**) Typical root system architecture displayed by 3-week-old *drb2* plants and the *drb2*/*MIR160B*, *drb2*/*mARF10* and *drb2*/*mARF16* transformant lines. Scale bar = 1.0 cm. Quantification of primary root length (**B**), lateral root number (**C**), and adventitious root number (**D**) in 3-week-old *drb2*, *drb2*/*MIR160B*, *drb2*/*mARF10* and *drb2*/*mARF16* plants. (**E**–**J**) RT-qPCR quantification of transcript abundance of *PRE-MIR160B* (**E**), miR160 (**F**), *eTM160-1* (**G**), *eTM160-2* (**H**), *ARF10* (**I**), and *ARF16* (**J**) in the roots of 3-week-old *drb2*, *drb2*/*MIR160B*, *drb2*/*mARF10* and *drb2*/*mARF16* plants. (**E**–**J**) Fold changes were determined by the 2^−∆∆CT^ method with the use of three biological replicates which contained six pooled plants per replicate. Averages of expression are represented as a fold change for each assessed transcript and were compared to the values obtained for *drb2* plants by a standard two-tailed *t*-test. Error bars represent the standard error of the mean (SEM) and asterisks show * *p* ≤ 0.05, ** *p* ≤ 0.01, *** *p* ≤ 0.001.

## Data Availability

All data reported here are available from the authors upon request.

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
