# Peer review of "Molecular Manipulation of the miR160/AUXIN RESPONSE FACTOR Expression Module Impacts Root Development in Arabidopsis thaliana"

_genes, 2024, doi:10.3390/genes15081042_

Round 1

Reviewer 1 Report

Comments and Suggestions for Authors

The article authored by Zimmerman et al. studied the functions of miR160/ARF module in root development in Arabidopsis thaliana, and highlighted the role of two DCL1 partner proteins, DRB1 and DRB2 in this process. The discoveries provide some new knowledge for the area of plant noncoding RNAs, but there are still some issues needed to be addressed by the authors.

Firstly, only one transgenic line was used in this study for each genotype.

Secondly, the most key point of this article is that the authors did not illustrate what is the main factor determining root development. The primary root length of three transgenic plants is promoted, but the expression levels of miR160, and the target gene ARF10,16,17 are different from each other. Although they claimed that the accumulation of miR160 was accounted for this result (line 27), actually the expression level of miR160 was not changed in Col-0/mARF16 shown in Fig 2F (not significant difference).

Thirdly, I could not conclude that which of the target genes was responsible for root development from their results. The results seemed very contradictory and no correlation between the metrics of root development (primary root length, lateral root number, adventitious root number) and the expression level of the target genes was found.

Fourthly, many long sentences were used and seemed very verbose, so that it is not easy for readers to read, and some qualifiers were not necessary. The presentation of the results seems tedious, and the authors should highlight the key points of the results with concise language.

Fifthly, a model should be proposed using a figure to tell the readers what is the main findings of this study to be clear in a glance.

Other issues:

The first sentence in the abstract is grammatically wrong, and no predicate was found in this sentence.

In line 181-182, they claimed that DRB2 is required for miR160 production from MIR160B precursor, however, Fig 1 showed that pre-MIR160B was downregulated in drb2 and in line 977-978, the authors admitted that pre-MIR160B was easier to be processed without DBR2. Therefore, the statement of line 181-182 is not right.

In line 187-190, the statements here were not consistent with Fig 1H, which showed the transcriptional miR160 was a bit higher in drb2 mutant. Thus, DBR2 did not have positive influence on miR160 accumulation.

In line 455-458, the author thought that the regulation of eTM160-1 and eTM160-2 by miR160 were translation repression like pattern, however, in line 660, they mentioned “miR160-directed cleavage-based silencing of the eTM160-2 transcript”. This is confusing and which one is right?

In line 885, the word “analyses” was not necessary.

In line 917 and1071, the authors quoted Fig S4H, but H was absent in Fig S4.

In line 917-920, the evidence was not sufficient to prove “DRB1 forms the primary DRB protein required to partner with DCL1 for efficient processing of this precursor transcript” merely by qPCR.

In line 998-1001, the authors could not conclude that ARF10 and ARF16 were regulated by miR160 via translational repression without determining the protein abundance of ARF10 and ARF16.

In line 1056-1058, they claimed that repression of ARF17 was the cause of lateral root promotion in Col/MIR160B, but how do they explained that in Col/mARF10, the lateral root number was not changed even ARF17 was also repressed?

Fig S3A, scale bar is missing.

Comments on the Quality of English Language

The English writing is not succinct enough to read, and some grammatical mistakes are contained (eg: the first sentence in the abstract). Please check carefully throughout the text.

Author Response

Reviewer #1

The article authored by Zimmerman et al. studied the functions of miR160/ARF module in root development in Arabidopsis thaliana, and highlighted the role of two DCL1 partner proteins, DRB1 and DRB2 in this process. The discoveries provide some new knowledge for the area of plant noncoding RNAs, but there are still some issues needed to be addressed by the authors.

Reviewer:         Firstly, only one transgenic line was used in this study for each genotype.

Response:        We have thoroughly outlined the process that we used in this study to identify the ‘best performing’ plant lines for each population of transformants. We have successfully used such an approach previously in numerous studies, including PMID: 33435199, PMID: 33396498, PMID: 33114207; and https://doi.org/10.3390/agronomy11091751

Reviewer:         Secondly, the most key point of this article is that the authors did not illustrate what is the main factor determining root development. The primary root length of three transgenic plants is promoted, but the expression levels of miR160, and the target gene ARF10,16,17 are different from each other. Although they claimed that the accumulation of miR160 was accounted for this result (line 27), actually the expression level of miR160 was not changed in Col-0/mARF16 shown in Fig 2F (not significant difference).

Response:        We thank the reviewer for identifying this oversight. We have thoroughly revised the text of all sections of the revised manuscript, and we clearly articulate the likely molecular cause of the root phenotypes displayed by some transformant lines.

Primary root elongation: Using the Col-0 transformant lines as an example, miR160 abundance is increased in Col-0/MIR160B, Col-0/mARF10 and Col-0/mARF16 transformants, which in turn enhances miR160-directed expression repression of ARF17. Via a similar mARF17 transgene approach, ARF17 has been previously shown to be a repressor of primary root elongation (PMID: 15829600). Therefore, decreased ARF17 transcript levels due to increased miR160 accumulation was the likely cause of primary root elongation in these three transformant lines. The molecular cause of the promotion of primary root length is also clearly explained in the revised manuscript.

Lateral root promotion: again, using the Col-0 transformant lines as an example, only the Col-0/MIR160B transformant showed promotion of lateral root development. In this line, miR160 abundance was increased and ARF17 expression was reduced. ARF17 is a known negative regulator of lateral root development (PMID: 15829600; PMID: 27292927), therefore reduced ARF17 transcript abundance in the roots of this transformant line was the likely cause of promoted lateral root development. Promotion of lateral root development in other transformant lines in the different genetic backgrounds is also explained in the revised manuscript.

Adventitious root development: the promotion of adventitious root development has also been clearly explained in the revised manuscript via the linkage of altered miR160, ARF10, ARF16 and ARF17 levels in transformant lines with promoted adventitious root development. We have also used previous studies (PMID: 15829600; PMID: 27292927; PMID: 19820192) to support our claims of the observed alterations to root system development.

Reviewer:         Thirdly, I could not conclude that which of the target genes was responsible for root development from their results. The results seemed very contradictory and no correlation between the metrics of root development (primary root length, lateral root number, adventitious root number) and the expression level of the target genes was found.

Response:        We thank the reviewer for identifying this oversight, and have addressed this concern of Reviewer #1 in the revised manuscript. Please see the above response outlining the explanation of the observed phenotypes in our revised manuscript, and the use of previous studies to support our claims (please see publications PMID: 15829600; PMID: 27292927; PMID: 19820192)

Reviewer:         Fourthly, many long sentences were used and seemed very verbose, so that it is not easy for readers to read, and some qualifiers were not necessary. The presentation of the results seems tedious, and the authors should highlight the key points of the results with concise language.

Response:        Again, we thank reviewer #1 for outlining this issue. We have dramatically reduced the overall page count (reduced by 6 pages) and the word count (reduced by over 3,000 words) of the revised version of our manuscript. We have concentrated on solely explaining the obtained results in the Results section, and have used the Discussion section of the revised manuscript to provide support of our findings. Further, we have attempted to remove all instances of verbose language, as well as to be much more concise in the written text of the revised manuscript.

Reviewer:         Fifthly, a model should be proposed using a figure to tell the readers what is the main findings of this study to be clear in a glance.

Response:        We now provide clear explanation of the findings presented in our study in the revised manuscript. Therefore, the inclusion of a Figure model does not seem necessary for reader understanding.

Other issues:

Reviewer:         The first sentence in the abstract is grammatically wrong, and no predicate was found in this sentence.

Response:        This has been addressed.

Reviewer:         In line 181-182, they claimed that DRB2 is required for miR160 production from MIR160B precursor, however, Fig 1 showed that pre-MIR160B was downregulated in drb2 and in line 977-978, the authors admitted that pre-MIR160B was easier to be processed without DBR2. Therefore, the statement of line 181-182 is not right. In line 187-190, the statements here were not consistent with Fig 1H, which showed the transcriptional miR160 was a bit higher in drb2 mutant. Thus, DBR2 did not have positive influence on miR160 accumulation.

Response:        In the revised manuscript we have carefully checked all instances of where the role of DRB2 in miR160 precursor transcript processing is discussed to ensure accuracy / consistency of statements made. The results presented in Figure 1H show that miR160 abundance is mildly elevated in drb2 roots due to the removal of DRB2 antagonism of DRB1 assisted processing of PRE-MIR160B (reduced abundance in Figure 1F), we therefore, fail to see relevance of much of this statement by the reviewer. Further, miR160 abundance does not have to be above or below some arbitrary determined threshold in order to be biologically relevant. Altered miR160 levels are obviously causing a phenotypic effect in the roots of the assessed Arabidopsis lines.

Reviewer:         In line 455-458, the author thought that the regulation of eTM160-1 and eTM160-2 by miR160 were translation repression like pattern, however, in line 660, they mentioned “miR160-directed cleavage-based silencing of the eTM160-2 transcript”. This is confusing and which one is right?

Response:        We thank Reviewer #1 for identifying this issue. All such references have been removed from the text of the revised manuscript.

Reviewer:         In line 885, the word “analyses” was not necessary.

Response:        Addressed.

Reviewer:         In line 917 and1071, the authors quoted Fig S4H, but H was absent in Fig S4.

Response:        The authors thank reviewer #1 for identifying this issue. All in-text references to Figure S4 have been corrected in the revised manuscript.

Reviewer:         In line 917-920, the evidence was not sufficient to prove “DRB1 forms the primary DRB protein required to partner with DCL1 for efficient processing of this precursor transcript” merely by qPCR.

Response:        We respectively disagree with this comment made by Reviewer #1. Repeat evidence to support this claim is provide through the manuscript.

Reviewer:         In line 998-1001, the authors could not conclude that ARF10 and ARF16 were regulated by miR160 via translational repression without determining the protein abundance of ARF10 and ARF16.

Response:        We agree with Reviewer #1 that we cannot definitely demonstrate translational repression without the use of protein analysis. We have therefore either removed or toned down such claims in the revised manuscript.

Reviewer:         In line 1056-1058, they claimed that repression of ARF17 was the cause of lateral root promotion in Col/MIR160B, but how do they explained that in Col/mARF10, the lateral root number was not changed even ARF17 was also repressed?

Response:        We have provide clear explanation of the molecular causes of the observed phenotypes in the transformant line series generated for assessment in this study in the revised manuscript. This issue has been addressed in the revised manuscript.

Reviewer:         Fig S3A, scale bar is missing.

Response:        We thank Reviewer #1 for identifying this oversight. We have corrected this issue in the revised manuscript.

Reviewer:         Comments on the Quality of English Language

The English writing is not succinct enough to read, and some grammatical mistakes are contained (eg: the first sentence in the abstract). Please check carefully throughout the text.

Response:        We have addressed all English language issue present in the original version of our manuscript. We thank reviewer #1 for their thorough and insight review of our manuscript. We are of the opinion that addressing the issues identified by Reviewer #1 has greatly improved the standard of our study.

Reviewer 2 Report

Comments and Suggestions for Authors

The research article titled “Molecular manipulation of the miR160/AUXIN RESPONSE FACTOR (ARF) expression module impacts root development in Arabidopsis thaliana” submitted by Kim et al. describing the microRNA160 (miR160) silencing signal for the regulation of the miR160 target genes, AUXIN RESPONSE FATOR10, 16 and 17, in various aspects of Arabidopsis development. Similarly, miR160 overexpression transgene (MIR160B), and two miR160-resistant transgene versions of ARF10 and ARF16 (mARF10 and mARF16) were developed and introduced in wild-type Arabidopsis plants, and the drb1 and drb2 mutant backgrounds defective in the activity of the core miRNA pathway machinery proteins,  DOUBLE STRANDED RNA BINDING1 (DRB1) and DRB2. This is a well-written article and I anticipate that the manuscript should be of great interest to the researchers working on plants functional genomics and plant biotechnology. I considered the manuscript needs the following minor improvements.

1.      The abstract section should be improved by indicating the implications of the findings, add some quantitative results and background information in the start of the abstract section.

2.      Introduction is too long and must be condensed and reader’s friendly. If this is a novel approach and not reported already you should add novelty in the introduction to increase its merit for citations.

3.      It is suggested to write down precisely the main objectives of the present work at the end of introduction.

4.      Divide section 2.2 into two sections (construction followed by line generation)

5.      The experiments were conducted in the green house kindly add information for growth condition photoperiod, humidity etc.

6.      Revise the statement Line 357-359, add another reference

7.      Add a conclusion section.  

8.      Phenotypic and molecular assessment of the miR160 expression module in the drb2 single mutant and the drb2/MIR160B, drb2/mARF10, and drb2/mARF16 transformant lines. The authors are suggested to add justification for the expression in case of (J) in the roots of 3-week-old drb2, drb2/MIR160B, drb2/mARF10 and drb2/mARF16 plants.

9.      There are some grammatical and typo mistakes, the MS should be thoroughly revised for English language for better understanding of the MS.

10.  Gene names/symbols should be italicized uniformly throughout the MS

Author Response

Reviewer #2

The research article titled “Molecular manipulation of the miR160/AUXIN RESPONSE FACTOR (ARF) expression module impacts root development in Arabidopsis thaliana” submitted by Kim et al. describing the microRNA160 (miR160) silencing signal for the regulation of the miR160 target genes, AUXIN RESPONSE FATOR10, 16 and 17, in various aspects of Arabidopsis development. Similarly, miR160 overexpression transgene (MIR160B), and two miR160-resistant transgene versions of ARF10 and ARF16 (mARF10 and mARF16) were developed and introduced in wild-type Arabidopsis plants, and the drb1 and drb2 mutant backgrounds defective in the activity of the core miRNA pathway machinery proteins,  DOUBLE STRANDED RNA BINDING1 (DRB1) and DRB2. This is a well-written article and I anticipate that the manuscript should be of great interest to the researchers working on plants functional genomics and plant biotechnology. I considered the manuscript needs the following minor improvements.

Dear reviewer,

We thank you for your thorough, constructive, and positive assessment of our manuscript. As part of the revision process, we have attempted to address all of your comments stemming from your review of our original submission. Please see our point-by-point responses below to each of your comments. Thank you again for identifying where in our manuscript improvements were required as part of the revision process, this was much appreciated.

Kind regards,

Andrew Eamens (on behalf of all authors)

Reviewer:    The abstract section should be improved by indicating the implications of the findings, add some quantitative results and background information in the start of the abstract section.

Response:   we thank the reviewer for identifying this oversight. We have completely reworked the Abstract to address this reviewer concern.

Reviewer:    Introduction is too long and must be condensed and reader’s friendly. If this is a novel approach and not reported already you should add novelty in the introduction to increase its merit for citations.

Response:   we thank Reviewer #2 for picking up this textual issue. We have rewritten much of the Introduction to (1) shorten its overall length, and (2) highlight the main focus of the study and the major findings of the study more clearly to the reader.

Reviewer:    It is suggested to write down precisely the main objectives of the present work at the end of introduction.

Response:   Please see the above response. We thank the Reviewer for identifying this issue and we have addressed this Reviewer concern in the revised manuscript. 

Reviewer:    Divide section 2.2 into two sections (construction followed by line generation).

Response:   Excellent suggestion by Reviewer #2, and actioned in the revised manuscript version.

Reviewer:    The experiments were conducted in the green house kindly add information for growth condition photoperiod, humidity etc.

Response:   full details of the growth conditions used for all Arabidopsis lines under analysis in this study were included in the original submission. We therefore have not addressed this reviewer comment.

Reviewer:    Revise the statement Line 357-359, add another reference

Response:   We thank Reviewer #2 for identifying this issue. The wording of all statements have been considered carefully as part of the revision process. Therefore addressed.

Reviewer:    Add a conclusion section.  

Response:   excellent suggestion by Reviewer #2. A Conclusion section has been added to the revised manuscript.

Reviewer:    Phenotypic and molecular assessment of the miR160 expression module in the drb2 single mutant and the drb2/MIR160B, drb2/mARF10, and drb2/mARF16 transformant lines. The authors are suggested to add justification for the expression in case of (J) in the roots of 3-week-old drb2, drb2/MIR160B, drb2/mARF10 and drb2/mARF16 plants.

Response:   in the thoroughly revised, and largely rewritten Discussion section of our revised manuscript we provide detailed molecular evidence of the likely cause of the phenotypes displayed by individual lines of the transformant series.

Reviewer:    There are some grammatical and typo mistakes, the MS should be thoroughly revised for English language for better understanding of the MS.

Response:   Thank you for pointing out these issues. As stated in the above response, much of the manuscript was rewritten to reduce the length of the manuscript by 6-pages, and ~3,000 words. We believe this major rewrite would have addressed such issues which existed in the original version of the manuscript while greatly improving both the flow and focus of the individual manuscript sections.

Reviewer:    Gene names/symbols should be italicized uniformly throughout the Manuscript.

Response:        As per the above, as part of the major rewrite undertaken during the revision process, we believe we have corrected such instances. We do thank you for bringing this issue to our attention however.

Round 2

Reviewer 1 Report

Comments and Suggestions for Authors

The manuscript is greatly improved after the revision. Most of the concerns have been addressed by the authors.

Comments on the Quality of English Language

The writing has been greatly improved and is much more easy to read than before.